# Study on the Spatial Allocation of Receding Land and Water Reduction under Water Resource Constraints in Arid Zones

Xin Yan [1,2], Yuejian Wang [1,2,*], Yuejiao Chen [1,2], Guang Yang [3,4], Baofei Xia [1] and Hailiang Xu [5]

[1] School of Science, Shihezi University, Shihezi 832000, China; 20202018007@stu.shzu.edu.cn (X.Y.); chenyuejiao@shzu.edu.cn (Y.C.); xiabaofei@shzu.edu.cn (B.X.)

[2] Key Laboratory of Oasis Towns and Mountain Basin System Ecology of Xinjiang Production and Construction Corps, Shihezi 832000, China

[3] College of Water and Architectural Engineering, Shihezi University, Shihezi 832000, China; yangguang@shzu.edu.cn

[4] Key Laboratory of Modern Water-Saving Irrigation of Xinjiang Production and Construction Corps, Shihezi 832000, China

[5] Xinjiang Institute of Ecology and Geography, Chinese Academy of Sciences, Urumqi 830011, China; xuhl@ms.xjb.ac.cn

[*] Correspondence: wyjian@shzu.edu.cn

**Abstract:** The withdrawal of cultivated land policy is not only an important task to promote cultivated land rest and alleviate the contradiction between supply and demand of water resources in arid areas, but also an important way to realize the sustainable development of agriculture and social economy. This study adopted the minimum per capita area method, ESPR (Exposure-Sensitivity-Pressure-Response) vulnerability assessment model, grey prediction model, and GIS spatial analysis. Furthermore, based on the characteristics of water resource constraints in the arid zone, Manas County was used as the study area. By exploring and analyzing the area of land retreat, through identifying its occurrence and position, the spatial zoning layout of land retreat can be realized to guarantee the effective implementation of water retreat and reduction. The following points were noted from the results: (1) the upper and lower limits of the area of receding land in Manas County were measured using the minimum per capita area method and the principle of balancing water supply and demand. The receding land in Manas County measured 16,493.68–20,749.90 hm$^2$, which accounted for 24.31–30.58% of the total area of cultivated land. (2) The results obtained from constructing the ESPR vulnerability assessment model, used to assess the vulnerability of cultivated land in Manas County, showed that the overall vulnerability of cultivated land in Manas County was high, with 94.74% of the county's cultivated land being moderately vulnerable or worse, which necessitates the optimization of land use. (3) The area of cultivated land withdrawal under the water resource constraint was used as a constraint for the withdrawal of cultivated land. Based on the evaluation of the vulnerability of cultivated land, with the results arranged from small to large, it was concluded that the area of cultivated land withdrawal in Manas County could reach up to 16,787.34 hm$^2$. There are four types of cultivated land withdrawals: desertified withdrawal, saline withdrawal, groundwater overexploitation withdrawal, and soil contamination withdrawal. The results of this study can provide a reference for Manas County to scientifically formulate a reasonable and orderly withdrawal system of farmland to reduce water use.

**Keywords:** arid zones; retreats for water reduction; spatial allocation of retreats; ESPR model



## 1. Introduction

The oasis is a unique and typical geographical feature of the arid zone. It provides opportunity for the survival of individuals and is an important center of economic activity in the arid zone. Consequently, its rise and fall are directly related to the evolution and development of the entire arid zone [1,2]. Water resources are essential for the development

of oases under arid climatic conditions and, thus, they determine the survival of oases [3]. In the past 50 years, the area of oases in Xinjiang has shown continuous expansion over a long period of time [4], from 36,900 km$^2$ to 147,600 km$^2$, with an annual expansion rate of 114.33 km$^2$/a [5,6]. In addition, the low water costs make the public less aware of water conservation.This led to the exploitation of water resources by more than 70%, and overexploitation in general [5]. Meanwhile, the imbalance in the configuration of water and soil in oases has made the disparity between water supply and demand increasingly more intense. This has had a direct impact on the livelihood of the population, economic development, and ecological security [2,7].

To guarantee the sustainable development of water resources, the No. 1 document (refers to the first programmatic document issued by the Central Committee of the Communist Party of China in 2011 on specific arrangements for accelerating water reform and development) of 2011 clearly stipulated that the most stringent water management system should be implemented, which is the "three systems" of total water use control, water use efficiency control, and limited pollution absorption in water function zones. Moreover, the "Three red lines" of total water use, water use efficiency, and water environment pollution control should be delineated [8]. With a per capita water supply of 4507 m$^3$, Xinjiang is the only province in China where water consumption exceeds the red line indicator for total water use control [9]. In Xinjiang, agriculture accounts for the highest proportion of water consumption, thereby crowding out ecological and other water uses, and increasing the risk of further deterioration of the ecological environment. To implement the water resources management system and reduce the proportion of agricultural land, the autonomous region carried out comprehensive work on efficient water conservation in agriculture. As a result, in 2016, the entire province of Xinjiang added 3.71 million hectares of highly efficient water-saving areas, with water-saving irrigation areas accounting for 55% of the total irrigation area [10]. Nevertheless, the amount of water saved currently does not meet the requirements of water reduction, thus, retreating cultivated land area has become an alternative way out of reducing agricultural water use. Therefore, it is of great practical significance to explore the balance of water resources and agricultural land in arid zone oases to solve the disparity between water supply and demand in the oases, maintain the ecological environment's regional water, and promote the sustainable development of the national economy and ecological civilization [1].

Studies on the ecological withdrawal of cultivated land locally and abroad have focused mainly on issues such as spatial and temporal differentiation or the landscape pattern evolution of the ecological withdrawal of cultivated land [11,12]. There have also been driving force studies [13,14], as well as studies evaluating how the issue impacts food security [15,16] and the facilitation of ecological compensation to farmers [17,18]. However, there are few studies on the withdrawal of cultivated land to reduce water resources utilization, which makes decision makers have no reference for which cultivated land should be withdrawn and how much cultivated land can be withdrawn. Therefore, the following study is undertaken to compensate for the withdrawal of cultivated land and to reduce water resources so as to provide a reference case for policymakers. Drawing on useful ideas such as spatial and temporal configuration and layout optimization of the ecological withdrawal of cultivated land, this study uses Manas County of the Manas Oasis in Xinjiang as the study area. Based on the Vulnerability Scoping Diagram (VSD) and Pressure-State-Response (PSR) models, we attempted to construct a new model for evaluating the vulnerability of the cropland-ESPR model, and then diagnose and identify the spatial distribution of cultivated land withdrawal. This study provides technical ideas and case references for the effective implementation of cultivated land withdrawal and water restoration projects.

## 2. Materials and Methods

### 2.1. Overview of the Study Area

Manas County is located in the hinterland of Xinjiang at a juncture between the northern foot of the Yilin-Habir Ga Mountains in the middle of the Tianshan Mountains (See Figure 1), the western end of the Changji Hui Autonomous Prefecture, and the southern edge of the Gurbantunggut Desert (43°28′29″ N–45°38′52″ N, 85°41′16″ E–86°43′10″ E). The climate belongs to the middle temperate continental arid and semi-arid climate, annual precipitation is 167 mm, annual average evaporation is 1195 mm, and evaporation is 11 times the precipitation. The surface water supply in Manas County mainly comes from Manas River, Taxi River and Qingshui River. The surface water supply is 2.192 billion cubic meters, accounting for 77% of the total water supply. The annual recharge of groundwater in the county is 405 million cubic meters. The recoverable reserves of groundwater are 230 million cubic meters, and the total amount of exploitation has reached 145 million cubic meters [6,7]. The county has a total area of 11,000 km$^2$, 13 towns and villages under its jurisdiction, and a population of 239,100 individuals as of 2019.

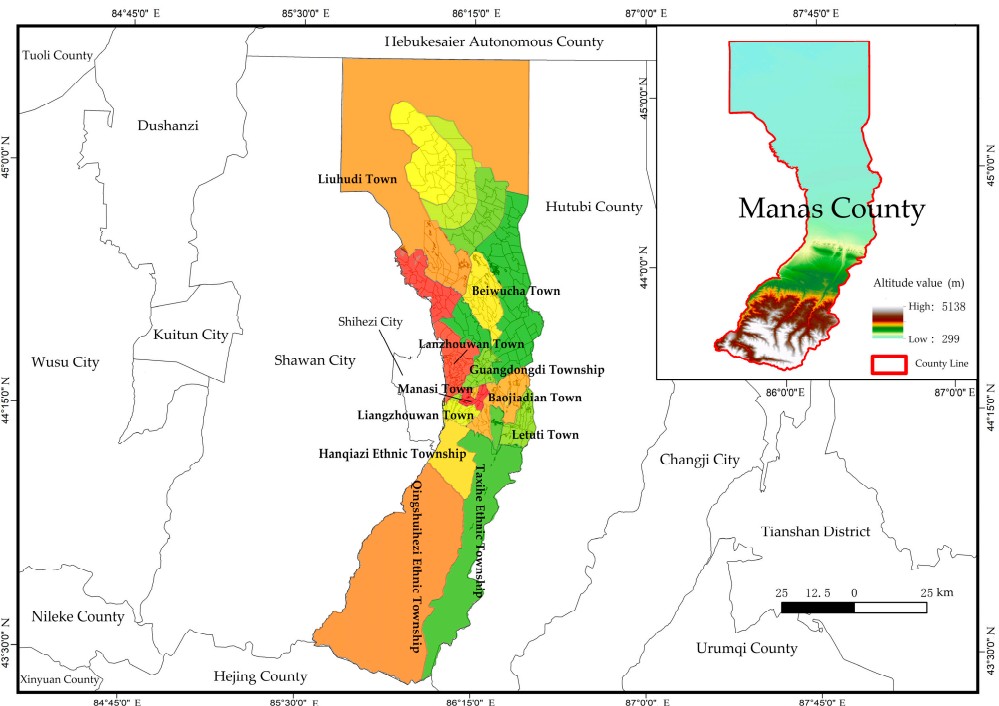

**Figure 1.** Spatial distribution of Manas County.

The topography of Manas County is high in the south and low in the north, and the landforms from south to north are mountainous areas, impact plains, and deserts, respectively. The rich vegetation and precipitation resources in the mountainous areas are the bases for the development of animal husbandry in the area. Furthermore, the main crop production areas are mostly in the impact plains in front of the mountains, where the soil is fertile and flat, accounting for 17.23% of the county's total land area. Owing to the natural and geographical environment of Manas County, watered land accounts for more than 99% of the total cultivated land [19], which shows that local agricultural production in this area relies heavily on water resources.

### 2.2. Data Sources

The data on cultivated land use in Manas County (including land-use maps; soil organic matter; soil salinity; irrigation and drainage; the areas under the use of chemical fertilizers, pesticides, and mulch; and land transfer areas) were obtained from the Manas County Agricultural and Rural Bureau. Socioeconomic data including population, grain

yield, replanting index (average number of crops planted on the same plot in one year), per capita income of farm households were obtained from the Manas County Statistical Yearbook and the Changji Prefecture Statistical Yearbook; meanwhile, soil erosion classification maps, annual average flow data of river course, and the soil erosion classification map and groundwater burial depth vector data were obtained from the Manas County Water Resources Bureau and the Ma Management Office. The distance data from rural settlements and roads were obtained with the help of the buffer zone analysis tool in ArcGIS 10.3, and the fragmentation of cultivated land was calculated using the landscape separateness index. The total amount of water resources limitation is based on the 'total water consumption control index' issued by Manas County, and the data of industrial water, urban public water, domestic water, and non-planting water are calculated according to the water scale and corresponding standards in the statistical yearbook.

*2.3. Research Methods*

The spatial allocation of cultivated land withdrawal mainly includes two aspects: (1) Calculations of the area of cultivated land withdrawal. As an important prerequisite for the implementation of land withdrawal and water reduction, food security must ensure that the area of cultivated land is necessary for regional food demand, and that the excess part is the area of cultivated land withdrawal. From the perspective of ecological security, adhering to the principle of water-fixed land, under the constraint of the total amount of water resources, the suitable area of cultivated land was calculated, and the area of cultivated land beyond that estimate was the area of cultivated land that may be withdrawn. (2) The identification of exited cultivated land. As a result of the harsh natural conditions and unsustainable levels of human activity in arid areas, the soil quality, fertility, and farming conditions of cultivated land in some areas have been degraded. By diagnosing the vulnerability of cultivated land, the "foothold" of exited cultivated land is identified. Thereafter, according to the main influencing factors of exited cultivated land, the exited cultivated land is divided to implement the return of the cultivated land.

2.3.1. The Area of Fallowing under Food Security

The total area of cultivated land under food security is measured by the minimum per capita area model. The area of retirement is equal to the current area of cultivated land minus the cultivated land holding. The minimum cultivated land per capita model is used to determine the cultivated land holding [15].

$$A = \frac{P \cdot F \cdot \alpha}{d \cdot l \cdot h} \tag{1}$$

A is the total area of cultivated land under food security in the study area, P is the per capita food demand in the target year, F is the population size in the target year, $\alpha$ is the food self-sufficiency rate in the target year, d is the yield per unit area of food in the target year, l is the food crop ratio in the target year, and h is the replanting index in the target year.

Due to the variability of cultivated land quality among regions, the standard coefficient of cultivated land productivity was used to correct for the total area of cultivated land under food security, and the formula was set as follows:

$$A_b = \frac{P \cdot F \cdot \alpha}{d \cdot l \cdot h} \cdot \frac{t \cdot T}{r \cdot R} \tag{2}$$

where $A_b$ denotes the revised the total area of cultivated land under food security in the study area; $t$ and $T$ denote the grain yields in the study area and the country, respectively; and $r$ and $R$ denote the grain-to-crop ratio in the study area and the country, respectively.

$$A_f = A_x - A_b \tag{3}$$

Af denotes the exited cultivated land area under food security, and $A_x$ denotes the area of current cultivated land in the study area.

2.3.2. The Area of Fallowing under Water Resource Constraints

(1) Water demand for cultivation

Water consumption in an area mainly comprises water for domestic use, water used on productive land, and ecological water. Water for production includes water for agriculture, industry, and urban public water; notably, water for agriculture also includes water for planting, forestry, and animal husbandry. Ecological water mainly refers to the ecological water in rivers [20,21]. The amount of water used for cultivation is then equal to total amount of water from the water resources minus the domestic, ecological, and non-cultivation water, as is seen in the equation below:

$$W_n = W - W_s - W_n^* - W_t \tag{4}$$

where $W$, $W_s$, $W_n$, $W_n^*$, and $W_t$ represent the regional water resources' red line, water for domestic use, water for planting and non-planting, and water for ecological use, respectively. Ecological water use was measured using the Tennant method to determine the water demand of the river [22], with the following formula:

$$W_t = \sum_1^{12} Q \cdot Z_i \tag{5}$$

$Q$ and $Z_i$ in the equation denote the multi-year average flow and baseflow percentage of the river, respectively. $Z_i$ taking values between 20% and 30% is considered to be the optimum water demand in an aquatic ecosystem [23].

(2) Area of Cultivated land Retirement under Water Resources Constraints

Based on the measurement for the water consumed by cultivation, the size of cultivated land was calculated by combining the crop cultivation structure and the irrigation quota of crops (the sum of irrigation quotas in the whole growth period of crops) [22]. It was then compared with the current size of cultivated land in the region. The excess represents the size of cultivated land that would need to be retired under water resource constraints. The formulae used are as follows:

$$A_w = A_x - A_c \tag{6}$$

$$A_c = W_n / I \tag{7}$$

where $A_w$ and $A_c$ are the area of the cultivated land withdrawal under water resource constraints and the area of suitable cultivated land under water resource constraints, respectively. I represents the irrigation quota for crops.

2.3.3. Identification of Fallow Lands

(1) Cultivated land vulnerability modeling

Determining which cultivated land needs to be retired should be guided by cultivated land use problems, combined with the region's natural, social, economic, and environmental indicators, to comprehensively diagnose the vulnerability of cultivated land and determine the level of urgency with which the land needs to be retired. In this way, the zoning layout of retired land can also be determined [24]. In terms of land system security evaluation, health evaluation (Evaluation on whether the land ecosystem can maintain normal operation), and vulnerability evaluation, the most widely used methods include the VSD and PSR models [24]. VSD model focuses on local and ecological conditions of land; PSR model focuses on land pressure and socio-economic factors, and as the two complement each other, they can reflect the overall situation of cultivated land. The VSD model consists of exposure (E), sensitivity (S), and adaptability (A) [25], whereas the PSR model consists of pressure (P), state (S), and response (R) [26]. The elements of the two

models influence each other. Moreover, by integrating and optimizing the two models and eliminating commonalities, an ESPR model consisting of exposure (E), sensitivity (S), pressure (P), and response (R) elements can be constructed (See Figure 2).

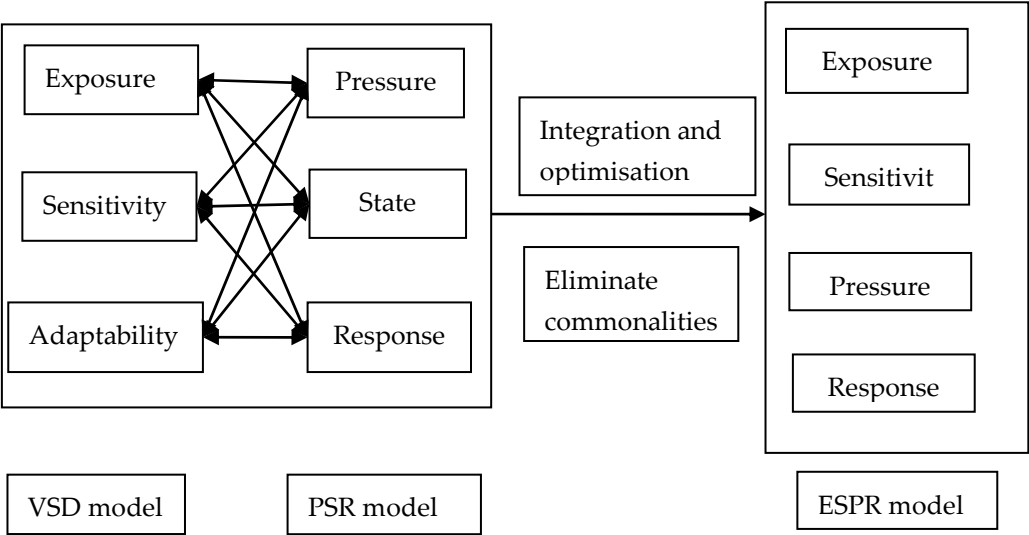

**Figure 2.** Construction of ESPR Model.

(2)　Selection of evaluation factors

The selection of evaluation factors should be based on the four elements of ESPR, in the context of the situation in Manas County, and compared with related studies [27–31]. Following the principles of scientificity, accessibility, and applicability, 22 indicators were selected from four aspects (natural, social, economic, and environmental) to construct an evaluation system to determine the vulnerability of the cropland ecosystem in Manas County, using the following model:

Exposure: Natural environmental factors were selected to reflect the natural local situation of the cultivated land. Topographic relief, soil salinity, soil organic matter content, total nitrogen, effective phosphorus, and fast-acting potassium were selected.

Sensitivity: This reflects the natural local presence of cultivated land due to human activities. Soil erosion intensity, cumulative variation in groundwater depth, agricultural film usage per unit area of cultivated land, pesticide usage per unit area of cultivated land, and fertilizer usage per unit area of cultivated land were selected.

Pressure: This reflects the intensification of the impact of human activities on cultivated land, and results in the use of cultivated land. The field size, cultivated land per capita, cultivated land fragmentation index, irrigation facility coverage, type of crops grown, and re-planting index were selected.

Response: While considering factors such as the natural background conditions of cultivated land, problems and directions of use, and characteristics of use, it is also important to consider socioeconomic factors and examine the impact of human activities on the system of cultivated land. The population density, distance of the plot from the settlement, distance from the main road, food output per unit area, and net income per farmer were selected.

Currently, the relevant bodies involved in the ecological and environmental assessment of the cultivated land withdrawal in China are still in the process of determining what to do; consequently, identifying local practices and expert advice is of great use. Therefore, the Analytic Hierarchy Process (AHP) method was applied to determine the weight of each evaluation factor and references were made to relevant studies [12,32–34]. Furthermore, the data were classified and normalized according to the observed data of the individual assessment indices in Manas County (See Table 1).

**Table 1.** Evaluation Index System of Cultivated Land Vulnerability.

| Target Layer | Criteria Layer | Indicator Layer | Meaning and Nature | Weighting |
|---|---|---|---|---|
| Fragility of arable ecosystems | Exposure (0.2161) | Topographic relief | The maximum relative elevation difference per unit area is a quantitative indicator to describe the landform. Positive | 0.0153 |
| | | Soil salinity | Serious salinization in the study area, selecting this index as an important and characteristic index of cultivated land vulnerability. Positive | 0.0784 |
| | | Soil organic matter content | The quality of cultivated land is characterized, which affects the growth of vegetation and further affects the safety of regional cultivated land ecosystem. Negative | 0.0591 |
| | | Total nitrogen | | 0.0211 |
| | | Effective phosphorus | | 0.0211 |
| | | Fast-acting potassium | | 0.0211 |
| | Sensitivity (0.2666) | Soil erosion intensity | The severe wind erosion in the study area led to the loss of nutrients on the surface of cultivated land and the low and unstable yield of cultivated land. This indicator was selected as an important and characteristic indicator of cultivated land vulnerability. Positive | 0.0936 |
| | | Cumulative variation in groundwater depth | Represents the degree of groundwater overexploitation. Positive value indicates decline of groundwater level, negative value indicates rise of groundwater level. Negative | 0.0404 |
| | | Film usage per unit area of cultivated land | Represents the non-point source pollution degree of cultivated land caused by agricultural film, chemical fertilizer, and pesticide. Positive | 0.0442 |
| | | Pesticide usage per unit area of cropland | | 0.0442 |
| | | Fertilizer usage per unit area of cultivated land | | 0.0442 |
| | Pressure (0.2886) | Field size | Field size affects the choice of returning farmland. The smaller the field size, the greater the possibility of exiting farmland. Negative | 0.0624 |
| | | Cultivated land area per capita | The smaller the per capita arable land, the stronger the dependence of farmers on arable land, the more vulnerable the livelihood of farmers. Negative | 0.038 |
| | | Cultivated land fragmentation index | The more broken the cultivated land is, the more broken the ecological environment is, the more prominent the contradiction between people and land is, and the pressure on the use of cultivated land is relatively large. Therefore, it is necessary to actively reduce human disturbance on cultivated land by exiting cultivated land. Positive | 0.0889 |
| | | Type of crops grown | Different types of crops, different intensity of cultivated land use | 0.0287 |
| | | Replanting index | Characterization of cultivated land use intensity. Positive | 0.0174 |

| Target Layer | Criteria Layer | Indicator Layer | Meaning and Nature | Weighting |
|---|---|---|---|---|
| Fragility of arable ecosystems | Pressure (0.2886) | Coverage of irrigation and drainage facilities | Cultivated land with good irrigation and drainage conditions should generally continue to be cultivated, play the role of farmland water conservancy facilities, and improve the utilization of farmland water conservancy facilities. | 0.0532 |
| | Response (0.2287) | Proportion of agricultural population | The larger the agricultural population, the larger the agricultural workforce that will be withdrawn from arable land and the greater the resistance to withdrawal. Negative | 0.0634 |
| | | Cultivated land transfer rate | The transfer of land can lead to complications in the distribution of subsidies to farmers who have withdrawn from farmland, increasing resistance to withdrawing from farmland. Negative | 0.028 |
| | | Per capita net income of farmers | The higher the returns received by farmers, the more dependent they are on agricultural production, and the more important cultivated land occupies in rural economic life, the greater will be the resistance to withdrawing from it. Negative | 0.0674 |
| | | Distance to residential areas | Generally distant and inaccessible arable land will have higher production costs and will be more likely to be withdrawn from cultivation land. Positive | 0.0346 |
| | | Distance from main roads | | 0.0353 |

The calculation method of comprehensive evaluation model of cultivated land ecosystem vulnerability is as follows [34]:

$$Z = \sum_{i=1}^{n}(Q_i \cdot Y_i) \qquad (8)$$

Z is the vulnerability of cultivated land ecosystem, that is, the urgency of exiting cultivated land. $Q_i$ is the weight of i index, $Y_i$ is the standardized value of i index. The urgent degree of withdrawal of cultivated land Z value range between [0, 1], the larger the Z value, indicating that the cultivated land is ecologically more unsafe, the stronger the urgency of withdrawal of cultivated land is, and withdrawal of cultivated land should be given priority; the smaller the Z value is, the smaller the ecological vulnerability of cultivated land plots is, and the lower the urgency of exiting cultivated land is, which can delay the exit of cultivated land.

(3)    Standardization of evaluation indicators

Before conducting a comprehensive evaluation, the classification criteria for qualitative and quantitative evaluation indicators must be determined, and the data should be processed without dimensions (See Table 2). Topographic relief and soil salinity were quantified by referring to relevant studies on the evaluation of cultivated land quality and stability; the soil organic matter content was quantified by referring to the County Agricultural Land Classification and Grading Regulations, and the results of the grading of cultivated land quality in Manas County [35]; the soil erosion classification was quantified by referring to the Soil Erosion Classification and Grading Standards, combined with relevant studies. Furthermore, the cumulative variation in groundwater depth was quantified by referring to the Manas River Basin and by referring to the study on the dynamic characteristics of groundwater level in the Manas River basin [36]; the load of

agricultural film per unit area of cultivated land, the load of pesticides per unit area of cultivated land, and the load of chemical fertilizers per unit area of cultivated land were quantified by referring to a study on the ecological risk evaluation of the surface pollution of cultivated land in Changji Prefecture [37]; the types of crops and replanting index were quantified according to the actual situation of crop sowing in Manas County; the radiation range of the impact of various factors on cultivated land was quantified according to the. The distance between rural settlements and major roads was quantified according to the actual situation of cultivated land use in Manas County. Additionally, the area of cultivated land and cultivated land fragmentation were quantified into five classes according to the data distribution characteristics of the observed values, and the specific quantification criteria are shown in Table 2. Proportion of agricultural population, land transfer situation, and farmers' per capita net income were taken from the statistical yearbook of Manas County. This data was standardized to the range of [0, 1] using the extreme value standardization method.

**Table 2.** Standardization of evaluation index.

| Indicator Layer | Quantitative Criteria for Indicators | | | | |
|---|---|---|---|---|---|
| | 0 | 0.25 | 0.5 | 0.75 | 1 |
| Degree of topographic relief (m) | >350 | 150~350 | 75~150 | 25~75 | 0~25 |
| Soil salinity (g/kg) | >7 | 6~7 | 4~6 | 2~4 | 0~2 |
| Soil organic matter content (g/kg) | <1 | 1~2 | 2~3 | 3~4 | >4 |
| Soil erosion intensity | Destructive | High Hazard | Hazardous | Low Hazard | Non-hazardous |
| Cumulative variation in groundwater depth of burial (m) | −3~−1 | −1~1 | 1~3 | 3~5 | >5 |
| Agricultural film usage per unit area of cultivated land (kg/km$^2$) | 295~300 | 290~295 | 285~290 | 285~280 | 280~275 |
| Pesticide usage per unit area of cultivated land (kg/km$^2$) | 0.57~0.82 | 0.39~0.57 | 0.23~0.39 | 0.12~0.23 | 0.03~0.12 |
| Fertilizer usage per unit area of cultivated land (t/km$^2$) | 7.99~10.02 | 5.87~7.99 | 3.24~5.87 | 1.32~3.24 | 0.21~1.32 |
| Type of crop planted | | Cotton-corn | | Wheat-rice | |
| Cultivation index | | 2 | | 1 | |
| Distance from residential areas (km) | >3.5 | 2.5—3.5 | 1.5~2.5 | 0.5~1.5 | <0.5 |
| Distance from main roads (km) | >4 | 3~4 | 2~3 | 1~2 | <1 |
| Field size (hm$^2$) | <1 | 1~3 | 3~5 | 5~7 | >7 |
| Cultivated land fragmentation index | >4 | 3~4 | 2~3 | 1~2 | <1 |
| Cultivated land transfer rate(%) | <5 | 5~10 | 10~15 | 15~20 | >20 |
| Cultivated land area per capita(People per mu) | <4 | 4~8 | 8~12 | 12~16 | >16 |
| Proportion of agricultural population (%) | <56 | 56~59 | 59~62 | 62~65 | >65 |
| Per capita net income of farmers (RMB) | <24,000 | 28,000~24,000 | 32,000~28,000 | 32,000~36,000 | >36,000 |
| Coverage rate of irrigation and drainage facilities | No Irrigation and drainage conditions | Partial irrigation and drainage | Irrigation and drainage conditions are generally met | Irrigation and drainage conditions are generally satisfactory | Sound irrigation and drainage facilities are available |

## 3. Results

### *3.1. Determination of the Area of Fallowing*

3.1.1. The Area of Fallowing from a Food Security Perspective

Important indicators of the food security capacity of a country or region are the per capita food requirements and the food self-sufficiency rate. Different values of per capita food demand will affect the amount of cultivated land held. Considering the National Medium- and Long-term Plan for Food Security (2008–2020) and information from related references [38–41], China's per capita food demand must be more than 425 kg/person. Xinjiang Province has a self-sufficiency rate of over 100%, but it is only in its own balance [39]. Therefore, with reference to national food security policies and relevant research results [39,42], the grain self-sufficiency rate in Manas County was set at 95%, which is basically self-sufficiency.

Based on the availability of data, the GM (1,1) grey prediction model was applied to predict the population size, grain yield, replanting index, and grain-to-crop ratio affecting food security in Manas County in the target year in this study, 2021. As can be seen from the table, the average multi-year relative error of each forecast value is within 5%, the equation fits well, and the forecast values are usable.

According to Equation (2), combined with the data in Table 1, the amount of cultivated land in Manas County in 2021 was calculated to be 47,096.77 hm². Based on the area of the existing cultivated land in Manas County, it can be estimated that the area of cultivated land that can be retired in 2021 for food security is 20,749.90 hm², which accounts for 30.58% of the county's cultivated land. Due to the limited data available, it was not possible to measure the population data, grain yield, replanting index, and grain-to-crop ratio of each township; therefore, the retired area was measured based on the proportion of cultivated land in each township (See Figure 3). Beiwucha Town had the highest area of cultivated land withdrawal, 4248.30 hm², which accounted for 20.47% of the total fallow area. This was followed by Baojiadian Town, with 3392.57 hm², which accounted for 16.35% of the total fallow area. The town with the least amount of cultivated land withdrawal was Qingshuihe Town, with only 384.37 hm². The area of cultivated land withdrawal determined from the perspective of food security is the upper limit of returnable cultivated land. This means that a maximum of 20,749.90 hm² can be returned, beyond which the demand for food will not be met, thus threatening food security.

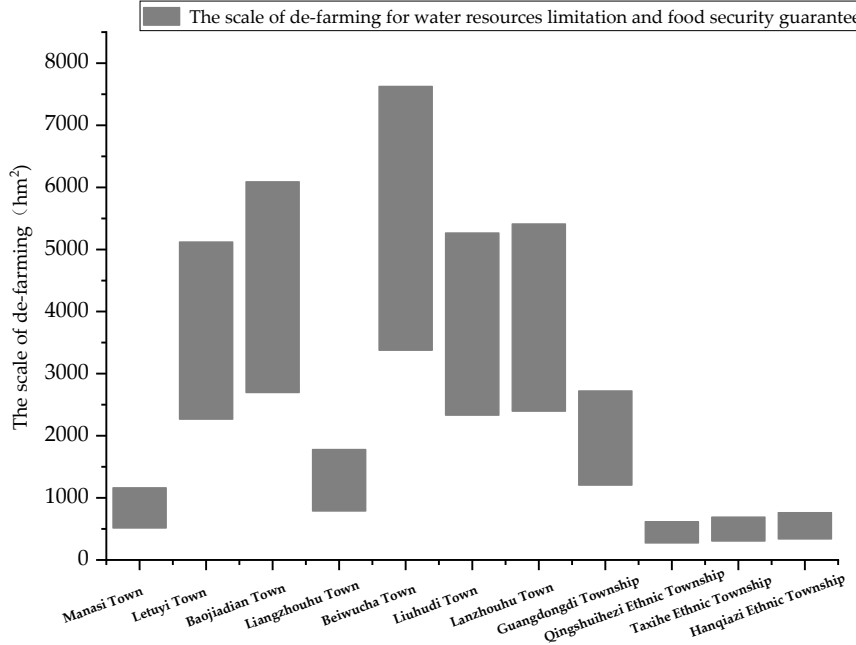

**Figure 3.** The area of de-farming for water resources limitation and food security guarantee.

### 3.1.2. The Area of Fallowing from a Water Limitation Perspective

By consulting the relevant statistical yearbooks of Manas County, combine the third formula, the amount of water used for cultivation in each township was measured (See Table 3). With water resource constraints, human needs are met first, followed by securing a certain amount of ecological water before it is eventually used for agricultural production. From the perspective of each township, due to the safeguarding of ecological water, Qingshuihe Town, Guangdongdi Town, and Liangzhouwan Town used less of the water that was allocated to the plantation industry, respectively. Meanwhile, Letuyi Town, Baojiadian Town, and Beiwucha Town had more water resources planned for themselves, thus, there was still a relatively large surplus of water for plantation production after other water uses were met.

**Table 3.** Prediction index of cultivated land possession.

| Forecast Indicators | Predicted Values | Multi-Year Mean Relative Error | Data Source |
|---|---|---|---|
| Population size (people) | 226,204 | 4.77% | 2007–2019 Statistical Yearbook of Changji Hui Autonomous Prefecture, Statistical Yearbook of Manas County |
| Grain yield (kg/mu) | 513 | 4.93% | |
| Replanting index (%) | 107% | 4.78% | |
| Grain to crop ratio (%) | 28.14% | 4.87% | |

Based on the calculated water consumption within the cultivation industry, combined with the irrigation quotas of the main crops grown in each township (See Table 4), it was calculated that the area of cultivated land under the water resource restriction was 51,352.99 hm². This means that the area of withdrawal of cultivated land was 16,493.68 hm², which accounted for 24.31% of the total cultivated area. The area of cultivated land that can be retired from water resources is the lower limit, which is the minimum area that can be retired. Any further reduction in the area of retired land will threaten ecological security.

**Table 4.** Water consumption of townships in Manas County.

| Township, Town | Total Planned Water Use (Million m³) | Water for Domestic Use (Million m³) | Non-Planting Water Use (Million m³) | Water for Cultivation (Million m³) |
|---|---|---|---|---|
| Manasi Town | 16.64 | 0.20 | 0.56 | 14.53 |
| Letuyi Town | 62.29 | 0.63 | 9.80 | 36.13 |
| Baojiadian Town | 66.88 | 0.70 | 13.65 | 36.80 |
| Liangzhoudu Town | 22.94 | 0.07 | 2.26 | 4.88 |
| Beiwucha Town | 71.35 | 0.18 | 15.47 | 39.97 |
| Liuhudi Town | 50.48 | 0.10 | 8.85 | 25.80 |
| Lanzhouwan Town | 32.49 | 0.13 | 2.53 | 14.10 |
| Guangdongdi Township | 23.34 | 0.12 | 2.87 | 4.62 |
| Qingshuihe Ethnic Township | 3.26 | 0.11 | 0.20 | 1.95 |
| Tashihe Ethnic Township | 7.26 | 0.03 | 0.45 | 5.43 |
| Hanqiazi Ethnic Township | 10.46 | 0.20 | 0.66 | 8.25 |

Under the dual conditions of water resource limitation and food security, the overall extent of the withdrawal of cultivated land in Manas County ranged from 16,493.68 hm² to 20,749.90 hm². As shown in Figure 3, the leading towns in Manas County were mainly Beiwucha Town, Baojiadian Town, Letuyi Town, Liuhudi Town, and Lanzhouwan Town, while other towns and villages took up less of the task of returning farmland to water.

### 3.2. Cultivated Land Vulnerability Assessment

The purpose of the cropland vulnerability assessment was to prioritize plots for the withdrawal of cultivated land. Based on the ESPR model, to calculate the comprehensive vulnerability score of cropland ecosystems, the equal spacing method was used to classify the comprehensive vulnerability score of cropland in the study area into five classes: extremely vulnerable ($0 \leq Z < 0.2$), severely vulnerable ($0.2 \leq Z < 0.4$), moderately vulnerable ($0.4 \leq Z < 0.6$), generally vulnerable ($0.6 \leq Z < 0.8$), and mildly vulnerable ($0.8 \leq Z < 1$).

According to the evaluation results for 14,340 cultivated land plots in Manas County, the situation of cultivated land use in this county is not optimistic (See Table 5). Based on the number of plots and the area of cultivated land, the majority of cultivated land plots were highly vulnerable, among which 424 plots were extremely vulnerable and need to be retired urgently. Moreover, 7716 plots were moderately vulnerable, which accounted for 87.62% of the area, and only 37 plots were mildly vulnerable. By comparing the observed data on cultivated land use conditions, it was found that 74.12% of the extremely vulnerable, severely vulnerable, and moderately vulnerable cultivated land was prone to soil erosion, and 75% of the cultivated land had a medium to high salinity rating. It can be seen that fragile resource and environment background and poor cultivated land use conditions lay the foundation of cultivated land ecosystem vulnerability in Manas County.

**Table 5.** Statistics of cultivated land vulnerability.

| Vulnerability of Cultivated Land | Number of Plots (pcs) | Number of Plots as a Percentage (%) | Cultivated Land Area (hm$^2$) | Cultivated Land Area as a Percentage (%) |
|---|---|---|---|---|
| Extremely vulnerable | 424 | 2.96 | 2175.57 | 3.21 |
| Severely vulnerable | 6024 | 42.34 | 2651.38 | 3.91 |
| Moderately vulnerable | 7716 | 53.81 | 59,450.05 | 87.62 |
| Generally vulnerable | 91 | 0.63 | 2501.98 | 3.69 |
| Mildly vulnerable | 37 | 0.26 | 1067.70 | 1.57 |
| Total | 14,340 | - | 67,846.67 | - |

Based on the spatial distribution (See Figure 4), these extremely fragile cultivated lands are mainly located in the northern part of Liuhudi Town and Beiwucha Town, the western part of Lanzhouwan Town, and the central part of other townships and surrounding areas. At the same time, fragmentation in these plots was high, the concentration was low, and the size of the plots was generally less than 5 hm$^2$. In addition, the depth of groundwater buried in these extremely vulnerable plots was relatively shallow, and there was a large amount of groundwater being extracted. The excessive input of pesticides, fertilizers, agricultural films, and other agricultural production materials into cultivated land year after year has greatly increased the land's vulnerability. The cultivated land located in the northern part of Manas County is closer to the desert, where the ecological environment is already fragile, and the interference of human activities makes the cultivated land extremely vulnerable. The cultivated land located in the western part of Lanzhouwan Town, the central part of other townships, and in the surrounding regions are in the built-up areas of the townships, which have a relatively concentrated population, frequent socioeconomic activities, and the construction of infrastructure such as cities and transportation. These factors not only destroy the form of cultivated land use and leads to a serious decline, but it also reduces the resistance to disturbance and the recovery capacity of the arable ecosystem, which has a very clear and negative impact on cultivated land.

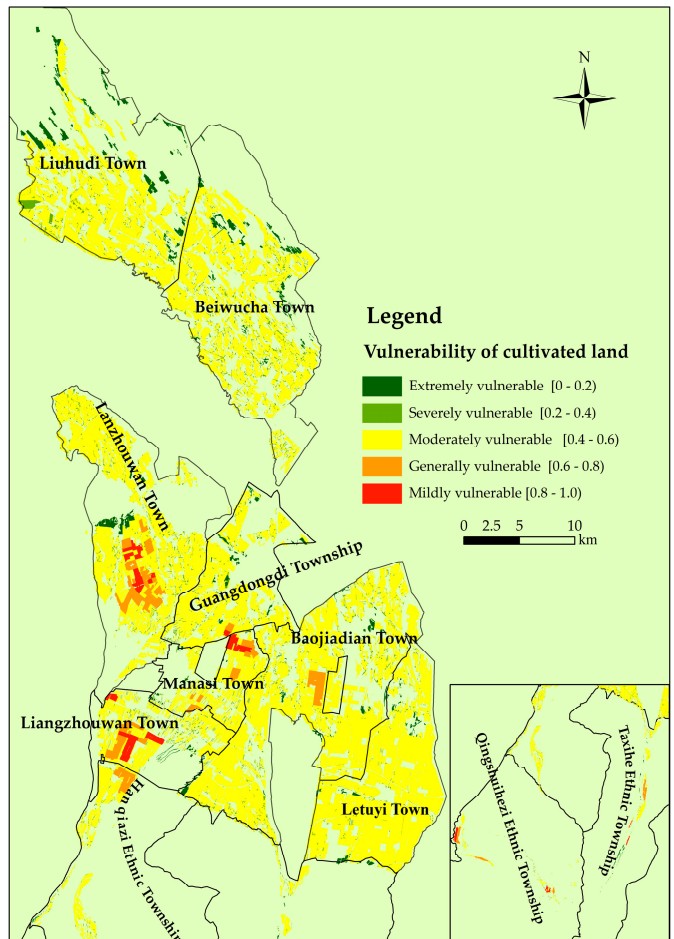

**Figure 4.** Spatial distribution of cultivated land vulnerability.

### 3.3. Identification and Layout of Fallow Land

　　Based on the evaluation results of cultivated land vulnerability as the basis for cultivated land withdrawal and using the area of cultivated land withdrawal under water resource restriction as the limit, the cultivated land parcels in Manas County were screened in reverse. This means that the cultivated land parcels were accumulated according to the value of the evaluation results of cultivated land vulnerability from smallest to largest; this was performed until the accumulated area was less than or equal to the area of withdrawal of cultivated land under water resource restriction. Using ArcGIS 10.3 (ESRI Corporation of the United States) to screen the cultivated land plots, it was concluded that the area of cultivated land that could be retired under water resource restrictions in Manas County in 2021 was 16,787.34 hm$^2$, which accounted for 24.74% of the cultivated land in the county and involved a total of 10,882 plots of cultivated land.

　　According to the spatial distribution of cultivated land withdrawal in Manas County, there are five clusters of cultivated land withdrawal (The dashed line in Figure 5), namely the northern part of Liutudi Town, the southern part of Beiwucha Town, the central part of Lanzhouwan Town, the northern part of Liangzhouhu Town, and the northern part of Letuyi Town (See Figure 5). Further analysis of the data, combined with the indicators that had a greater impact on the vulnerability score of cultivated land, led to the identification of four main types of withdrawal of cultivated land: desertification withdrawal, salinization withdrawal, severe overextraction of groundwater withdrawal, and soil pollution withdrawal. The northern edge of Liuhudi Town and Beiwucha Town is adjacent to the desert, and has a poor ecological environment, poor cultivation conditions, and comprises low-yielding cultivated land without irrigation and drainage facilities. Consequently, this area faces the risk of desertification.

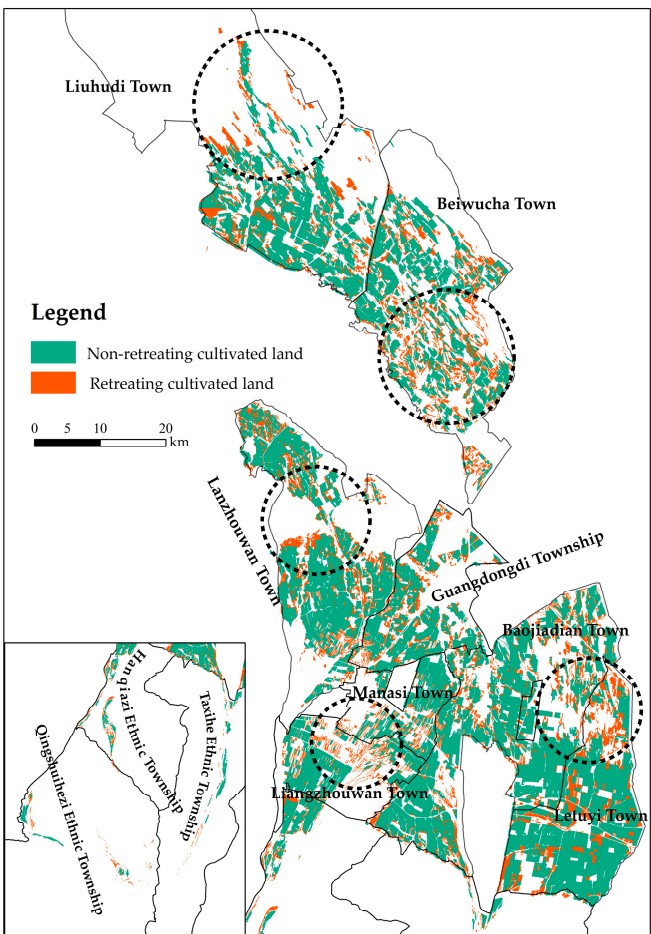

**Figure 5.** Spatial distribution of cultivated land in Manas County.

The salt content of the soil in the southern part of Beiwucha Town and the central part of Liangzhouhu Town was above 4 g/kg, which is moderately saline and seriously affects the growth and development of crops, even cotton, which is a salt-tolerant crop. As a result, the income per unit area of the farmers in this area was low. This type of cultivated land withdrawal is, therefore, salinization.

Lanzhouwan Town has a high level of mechanized agricultural production. Furthermore, the amount of pesticides and fertilizers used on its cultivated land, and the area covered by mulch, were the highest in Manas County, making the soil highly polluted by pesticides, fertilizers, and mulch. Meanwhile, the Jiehezi Reservoir in Lanzhouwan Town, which can irrigate 133,000 ha of cultivated land, is the main source of irrigation for the surrounding cultivated land [43]. To avoid the impact on the water quality of the reservoir, all the surrounding cultivated land is fallowed. This type of cultivated land withdrawal is due to soil pollution.

The cultivated land in the northern part of Letuyi Town and the eastern part of Baojiadian Town had the largest cumulative variation in groundwater depth, and all of them had negative variations, ranging from $-3$ m to $-2$ m. Compared to the current distribution of machine wells in Manas County, the cultivated land in the northern part of Letuyi Town and the eastern part of Baojiadian Town have more dense machine wells, which makes the depth of groundwater in this area much higher than that in other areas, and there is a very serious phenomenon of groundwater overexploitation. It is therefore necessary not only to withdraw a certain amount of cultivated land, but also to remediate and bury some of the illegally dug wells to restore the depth of groundwater. The withdrawal of cultivated land in this area is due to a serious case of groundwater overexploitation.

## 4. Discussion

This study first estimated the upper and lower limits of the withdrawal of cultivated land area in Manas County from the perspectives of food security and water resources constraints, which was an important research finding. On this basis, the vulnerability evaluation model of cultivated land was constructed to identify the specific location of the withdrawal of cultivated land, which provided technical reference for the implementation of the policy of reducing water resources utilization in the withdrawal of cultivated land, but there are still some deficiencies that will be the direction of future research.

The area of cultivated land withdrawal is not always constant. From the perspective of food security, an increase in the value of a parameter such as food productivity, grain ratio or multiple cropping index may increase the area of cultivated land withdrawal; from the perspective of water resource constraints, under the total amount control, the amount of water used for planting production is determined by domestic water, ecological water, and other non-planting water. When the water used in the above three aspects is satisfied, there are still more water resources for planting production, and the area of cultivated land withdrawn may be reduced. However, the amount of cultivated land increased or decreased still needs to be accurately calculated. Therefore, in future research, a dynamic evaluation platform for cultivated land withdrawal will be established to meet the needs of changing parameters.

The area of exited farmland is not a one-time all exit, need to be differentiated, it would also be necessary to carry out differentiated cultivated land withdrawals, dividing the exiting cultivated land into permanently exiting cultivated land and precariously exiting cultivated land. For some areas where the vulnerability of cultivated land is serious, the management of permanently withdrawn cultivated land is implemented, and these cultivated lands are managed to prevent their degradation. For some areas with good farming conditions, cultivated land where local natural and climatic conditions have improved can be withdrawn from cultivation for a short period of time, and recultivation can be considered when it is restored to a cultivated state.

In the selection of cultivated land vulnerability evaluation factors, the characteristics of cultivated land use and environmental differences were highlighted, while the attribute of cultivated land was less considered. For example, priority should be given to withdrawal of cultivated land after the expiry of its contractual life, and priority should also be given to withdrawal of cultivated land that is illegally reclaimed, which includes cultivated land that is not on the Land Use Status Map of the Ministry of Land and Resources. In future studies, the land contract rights database and the land use status map can be superimposed on the cultivated land vulnerability distribution map to consider the withdrawal issue in an integrated manner. In addition, withdrawal from cultivated land has the greatest impact on farmers' livelihoods, and this study does not take the effect of farmers' willingness to withdraw on land use into consideration. Therefore, in future cultivated withdrawal operations in other regions, the spatial distribution of fallowed land can be explored by combining factors such as the farmers' willingness and livelihood transition.

After withdrawing from cultivated land, how to resettle the remaining labor is an important issue. For farmers with permanently withdrawn cultivated land, apart from giving some compensation for withdrawal, decision makers can also guide the remaining laborers to carry out industrial transformation and engage in secondary and tertiary industries. Only in this way can we implement the policy of withdrawing from cultivated land and reducing the use of water resources in an orderly manner, so as to ensure that the withdrawn cultivated land is no longer cultivated. For farmers with short-term farmland withdrawal, decision makers can only give compensation amounts during the period when the cultivated land is withdrawn and require that it be managed and protected. In addition, a reward system can be established to increase farmers' motivation to protect their cultivated land.

## 5. Conclusions

Based on the contradictory relationship between water scarcity and cultivated land use in arid areas, this study took Manas County as the study area and empirically explored the spatial allocation of cultivated land withdrawal in the region. This was done using the minimum area per capita method, ESRP vulnerability assessment model, grey prediction model, GIS spatial analysis, and other methods, to provide reference for the effective implementation of the fallow water reduction policy. From this study, the following conclusions were drawn:

(1) Through the minimum area method and the principle of the balance between water supply and demand, the lower and upper limits of withdrawal from cultivated land in Manas County were measured to be 16,493.68–20,749.90 hm$^2$, which accounted for 24.31–30.58% of the total area of cultivated land, respectively.

(2) The vulnerability of cultivated land in Manas County was evaluated using the ESRP vulnerability assessment model. The overall vulnerability of cultivated land in Manas County is high, with 94.74% of the cultivated land in the county being moderately vulnerable or worse. Furthermore, there is an urgent need to optimize the way cultivated land is used.

(3) Using the area of cultivated land withdrawal under water resource constraints as a constraint for the withdrawal of cultivated land and arranging them from small to large according to the results of the vulnerability evaluation of cultivated land, the area of withdrawal of cultivated land in Manas County totaled 16,787.34 hm$^2$, which accounted for 24.74% of the county's cultivated land area. Moreover, there were five spatial aggregations of fallowed land and four types of withdrawal of cultivated land. The four types of withdrawal of cultivated land were desertification withdrawal, salinization withdrawal, severe overextraction of groundwater withdrawal, and soil pollution withdrawal.

**Author Contributions:** Conceptualization, X.Y.; methodology, Y.W.; software, H.X.; validation, X.Y.; formal analysis, B.X.; investigation, X.Y.; resources, Y.W.; data curation, X.Y.; writing—original draft preparation, X.Y.; writing—review and editing, Y.C.; visualization, G.Y.; supervision, Y.C.; project administration, Y.W.; funding acquisition, Y.W. All authors have read and agreed to the published version of the manuscript.

**Funding:** This research was funded by the Shihezi University Youth Innovation and Cultivation Talent Program Project (Project Numbers: KX00300302), Shihezi University's Innovative Development Special Project (Project Numbers: CXFZSK202105), Shihezi University High-level Talents Research Start Project (Project Numbers: RCZK2018C41 and RCZK2018C22), and Corps key areas of science and technology research program projects (Project Number: 2021AB021).

**Institutional Review Board Statement:** Not applicable.

**Informed Consent Statement:** Not applicable.

**Conflicts of Interest:** The authors declare no conflict of interest.

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
