# Peer review of "Study on the Spatial Allocation of Receding Land and Water Reduction under Water Resource Constraints in Arid Zones"

_agriculture, doi:10.3390/agriculture12070926_

Round 1

Reviewer 1 Report

The study deals with agricultural production in a water limited area of northern China. It undertakes an estimation of the area that can be withdrawn from agriculture without jeopardizing local food security and the area that should be withdrawn to secure the available water resources. It proceeds with locating the areas that should be prioritized for this withdrawal according to the vulnerability of agricultural activities. The study is interesting and the concept is valid. The overall value however is affected by the sometimes vague description of methods, data and assumptions and the limited discussion of the findings.

Concerning the language and although I am not a native speaker of english I feel that the MS needs to be rechecked. I found the use of past tense especially confusing, in many instances I got the impression that land is already withdrawn from agriculture but the wider context of the text suggested that the authors actually meant that land should be withdrawn from agriculture according to their study results. In a few instances the meaning of a phrase is unclear or a phrase is broken abruptly without being completed (e.g. ‘The "Three red lines" [8]’ in line 59, lines 254-255). Also abbreviations are not spelled out in their first use and/or included in the abstract without explanation (eg. ESPR, AHP, VSD, PSR).

Specific comments on the MS sections:

Introduction

l. 54-55 It is unclear what ‘the No. 1 document of 2011’ refers to.

l. 67 ‘metric units’ which units are referred to?

Materials and Methods

The MS is about water saving but no precipitation data or aridity index values are presented. Some information about spatial variability of the climate within the study area as well as interannual variability of weather conditions would be helpful to understand the context of the study.

In line 130 ‘the scale and the “quantity” of cultivated land withdrawal’ appears to be a single value referring to area. I suggest that the authors should use the term ‘area’ for this purpose consistently throughout the MS.

Subsection 2.3.1 is especially problematic. I suggest that the authors include an appendix fully explaining each variable, what it represents and the assumptions underlying it. I also cannot understand the meaning of some expressions. What is the ‘cultivated land holding’ or the ‘replanting index’ in this context?

In subsection 2.3.2 a method is presented for estimating the amount of water available for agriculture. However in the data sources section there is no reference to data about the the total available amount of water and water discharge by river flow that are essential elements of the presented equation.

In subsection 2.3.3 I have serious doubts whether the indicators listed as response indicators can be accepted as such, except from ‘land transfer rate’. E.g. in which sense is ‘distance from main roads’ a response? Overall all these indices are useful but they are not responses. True responses are the ones described in the introduction section in lines 61-66 and the compensation mentioned in line 79. Again the meaning of ‘film load’ (e.g. table 2) and ‘health evaluation’ (line 195) is not clear.

With respect to table 2 not all the variables mentioned in table 1 are listed, like population density, land transfer situation, and farmers' per capita net income. I suspect that ‘ land transfer situation’ equals ‘Concentration index of cultivated land’ but this should be made explicit. Still the range of values for each indicator layer for the missing variables must be included. In section 3.2 a composite score is presented, however the way it is calculated is not presented. The authors should present the method used in the materials and methods section.

Results and Discussion

Section 3 ‘Results and Discussion’ should be renamed to plain ‘Results’ since a Discussion section follows.

In many cases the information presented in tables or figures is repeated in the text, e.g in lines 291-295, 305-308, a practice that should be avoided. In section 3.2 the text refers to ‘patches’ ans table 5 to ‘plots’. Are these terms used interchangeably? If yes, choose one of them and use consistently; if no the relationship between them must be explained.

In line 391 the expression ‘poor ecological conditions’ must be rephrased. The conditions may be favorable for agricultural production or not but in an ecological sense they are neither poor or rich, such value judgments are meaningless when referring to ecology.

Discussion

The discussion is very poor. It is limited to suggestions for future research and does not discuss the findings. Lines 418-443 should be formulated more concisely and form the end of the section. I consider the finding that the upper limit of land fallowing in order to ensure local food security is compatible with the lower limit needed to ensure water use sustainability encouraging and the most important finding. The authors should also discuss their assumptions about the values of the variables used in their analysis; how sensitive is this finding if these assumptions are modified? E.g what about if water saving measures covered more cultivated area?

I also suggest that two issues should be mentioned. Firstly, apart from regional food security the current excess current agricultural production will probably be missed from elsewhere if land is withdrawn from agriculture. What are the implications? Can they hinder the implementation of a land withdrawal policy? Secondly and if the interannual weather variability is large what are the prospects of adopting a flexible approach cultivating more land in favorable years?

Further the expression ‘theoretical guidance’ in line 419 is misleading, the authors present no theory or relevant guidance upon which a policy can be based. Instead they present practical suggestions.

Conclusions

The methods do not need to be repeated in the conclusion section.

In line 453 the order of mentioning the upper and lower limits is apparently reversed. It should be corrected.

Reviewer 2 Report

The manuscript entitled “Study on the spatial allocation of receding land and water reduction under water resource constraints in arid zonesaimed to provide technical ideas and case references for the effective implementation of cultivated land withdrawal and water reduction policies by using minimum per capita area method, ESPR vulnerability assessment model, grey prediction model, and GIS spatial analysis. Though the paper is well written but can be improved significantly as follows:

·         Abstract can be improved on a qualitative basis.

·         Introduction can be improved by linking the research gap and research objectives. There is a need to explain the motivation for the methodology used.

·         In section 2.2 try to provide the reference of the data source, if applicable. Try to correlate food security with water resources.

·         For the case of equations, try to give more investigative parameters that conceptualize and enhance the model understanding.

·          What is the total water demand for the study area and the shortfall of the water supply?

·         Full form of the model, i.e., VSD needs to be mentioned for the general readers' clarity.

·         What does the nature of the indicator i.e., positive, or negative show, and how does it affects the cultivated land vulnerability?

·         Try to discuss results on a scientific basis instead of spatial characteristics of the study area.

·         Do not use the “X” in all the equations to represent multiplication

·         Headings names to be updated accordingly “3. Results and Discussion” and “4. Discussion”.

Round 2

Reviewer 1 Report

The authors have addressed the comments made in the first review. A few additional issues should be considered:

lines 100-107 Add the source of information on water resources

lines 236-241 While I see the rational of choosing these variables to indicate “responses” following the additional information included in table 1, still they are not responses. The authors should slightly rephrase their section making clear that the chosen variables affect or shape the responses and are not responses themselves.

Figure 1 The legend of the map in the upper right corner has no reference to the presented attribute or units. It seems that it refers to altitude. Reference must be made either in the legend or the caption of the figure, including the units.

Avoid numbering in discussion sections.

A few language issues persist.

Line 48-49. “In addition, the low water costs and the weak water-saving awareness of the masses.” The verb is missing, the meaning is not clear.

Line 76 “water ecological civilization”. There is no such term. Express more clearly.

Lines 84-85 “Therefore, in or-der to make up for the withdrawal of cultivated land to reduce water resources research.” Something is missing.
